# Reshaping of bacterial molecular hydrogen metabolism contributes to the outgrowth of commensal *E. coli* during gut inflammation

Elizabeth R Hughes[1], Maria G Winter[1], Laice Alves da Silva[2], Matthew K Muramatsu[1], Angel G Jimenez[1], Caroline C Gillis[1†], Luisella Spiga[1‡], Rachael B Chanin[1], Renato L Santos[2], Wenhan Zhu[1], Sebastian E Winter[1,3]*

[1]Department of Microbiology, UT Southwestern, Dallas, United States; [2]Departamento de Clinica e Cirurgia Veterinarias, Escola de Veterinaria, Universidade Federal de Minas Gerais, Belo Horizonte, Brazil; [3]Department of Immunology, UT Southwestern, Dallas, United States

**ABSTRACT** The composition of gut-associated microbial communities changes during intestinal inflammation, including an expansion of Enterobacteriaceae populations. The mechanisms underlying microbiota changes during inflammation are incompletely understood. Here, we analyzed previously published metagenomic datasets with a focus on microbial hydrogen metabolism. The bacterial genomes in the inflamed murine gut and in patients with inflammatory bowel disease contained more genes encoding predicted hydrogen-utilizing hydrogenases compared to communities found under non-inflamed conditions. To validate these findings, we investigated hydrogen metabolism of *Escherichia coli*, a representative Enterobacteriaceae, in mouse models of colitis. *E. coli* mutants lacking hydrogenase-1 and hydrogenase-2 displayed decreased fitness during colonization of the inflamed cecum and colon. Utilization of molecular hydrogen was in part dependent on respiration of inflammation-derived electron acceptors. This work highlights the contribution of hydrogenases to alterations of the gut microbiota in the context of non-infectious colitis.

*For correspondence:
sebastian.winter@
utsouthwestern.edu

Present address: †Genentech Inc, South San Francisco, United States; ‡Department of Pathology, Microbiology, and Immunology, Vanderbilt University Medical Center, Nashville, United States

## Introduction

The mammalian distal gut is densely colonized by a community of bacteria, archaea, viruses, and eukaryotic microorganisms, collectively termed the gut microbiota. Under homeostatic conditions, the gut microbiota is dominated by obligate anaerobic bacteria in the Bacteroidetes and Firmicutes phyla (*Eckburg, 2005*; *Moore and Holdeman, 1974*). Members of the phyla Actinobacteria, Verrucomicrobia, and Proteobacteria constitute minor populations in the healthy gut microbiota. Obligate anaerobic bacteria successfully colonize the healthy gut due to their ability to degrade the available complex polysaccharides (*Kaoutari et al., 2013*) (reviewed in *Cockburn and Koropatkin, 2016*; *Koropatkin, 2012*) and their ability to thrive in an anaerobic environment through fermentation (*Hartman, 2009*; *Jalili-Firoozinezhad, 2019*; *Litvak et al., 2018*).

During intestinal inflammation, the composition of the gut microbiota changes compared to the homeostatic state. Microbial diversity decreases (*Manichanh, 2006*; *Mirsepasi-Lauridsen, 2018*), the prevalence of mucolytic bacteria in the mucosa increases (*Png, 2010*), and populations of Enterobacteriaceae family members expand (*Haberman, 2014*; *Kotlowski, 2007*; *Lupp, 2007*). Disease-associated changes in gut microbiota composition have been observed in patients with inflammatory bowel disease (IBD) (*Frank, 2007*; *Kotlowski, 2007*), enteric pathogen infection (*Lupp, 2007*;

*Stecher et al., 2007*; *Wang et al., 2019*), necrotizing enterocolitis (*Mai et al., 2011*), and in animal models of colitis (*Garrett, 2007*; *Lupp, 2007*). Experiments in animal models suggest that mucosal host responses contribute to microbiota changes (*Winter and Bäumler, 2014*; *Winter, 2013*) and, conversely, microbial communities can instigate or perpetuate disease in the context of genetic susceptibility (*Garrett, 2010*; *Manichanh, 2012*; *Zhu, 2018*).

In IBD patients, disease-associated changes in the microbiota composition, genetic coding capacity, and fecal metabolite concentrations not only correlate (*Franzosa, 2019*), but the availability of certain metabolites impacts microbial community structure (*Fornelos, 2020*; *Hughes, 2017*). The inflammatory response produces reactive oxygen and nitrogen species, which when degraded in the gut lumen produce the electron acceptors tetrathionate, nitrate, and oxygen (*Chanin, 2020*; *Winter, 2010*; *Winter, 2013*). Additionally, changes in colonocyte metabolism during inflammation also result in an increased availability of the electron acceptor oxygen (*Cevallos, 2019*; *Hughes, 2017*; *Litvak, 2019*; *Lopez, 2016*; *Rivera-Chavez, 2016*). Findings from murine models of colitis suggest that the metabolic versatility of Enterobacteriaceae, particularly the ability to utilize a large repertoire of terminal electron acceptors, allows Enterobacteriaceae family members to thrive in the inflamed gut. Enterobacteriaceae utilize these inflammation-derived electron acceptors to perform anaerobic and aerobic respiration in the inflamed gut, thus gaining a fitness advantage over bacteria that rely on fermentation, a less energetically favorable metabolism. A functioning electron transport chain also enables utilization of poorly accessible carbon sources (*Faber et al., 2017*; *Fornelos, 2020*; *Price-Carter, 2001*; *Spiga and Winter, 2019*).

The role of respiratory dehydrogenases in inflammation-associated outgrowth of Enterobacteriaceae during non-infectious colitis is underexplored. A functioning electron transport chain requires oxidation of an electron donor by a respiratory dehydrogenase, shuttling liberated electrons to the quinone pool, and reduction of an electron acceptor via terminal reductases and oxidases. Reduction potentials and gene regulation determine which combinations of electron-donating and -accepting reactions occur under physiological conditions (*Unden et al., 2014*). Formate dehydrogenases contribute to the expansion of Enterobacteriaceae in murine models of IBD (*Hughes, 2017*). Under laboratory conditions, Enterobacteriaceae utilize several other molecules as electron donors, such as molecular hydrogen ($H_2$). Therefore, we focused on investigating $H_2$ metabolism in the inflamed gut in the current study.

Hydrogenases are a diverse family of metalloenzymes that catalyze the oxidation and/or production of molecular hydrogen (*Benoit et al., 2020*; *Pinske and Sawers, 2016*). These enzymes are typically classified based on the metal content of the active site and their biochemical activity (*Peters, 1998*; *Shima, 2008*; *Anne, 1995*). The active site of [NiFe]-hydrogenases contains nickel and iron, while the active site of [Fe]-hydrogenases and [FeFe]-hydrogenases includes one or two iron atoms, respectively. Based on their activity, hydrogenases can be further categorized into uptake, evolving, bidirectional, bifurcating, and sensory enzymes (*Greening, 2016*; *Vignais and Billoud, 2007*). Uptake hydrogenases convert $H_2$ to two protons and two electrons, with the two electrons often participating in an electron transport chain. Conversely, hydrogenases defined as evolving are responsible for production of $H_2$. Bidirectional hydrogenases can produce or oxidize $H_2$ (reviewed in *Tamagnini, 2007*). Bifurcating hydrogenases are enzymes involved in $H_2$ metabolism that use an exergonic reaction (e.g., oxidation of ferredoxin) to drive an endergonic reaction (e.g., oxidation of NADH) without an ion gradient (*Li, 2008*; *Schut and Adams, 2009*). Electron bifurcation was only recently demonstrated, yet the genomes of many anaerobes encode predicted bifurcating hydrogenases (*Greening, 2016*; *Schut and Adams, 2009*). Sensory hydrogenases detect changes in $H_2$ partial pressure and then activate regulatory cascades controlling expression of additional hydrogenases (*Lenz and Friedrich, 1998*), but additional study is required to characterize the many putative sensory hydrogenases present in obligate anaerobes (*Greening, 2016*).

Microbial $H_2$ metabolism in the inflamed gut is incompletely understood. Here, we mined previously published shotgun metagenomic sequencing datasets of human IBD patients and healthy controls (*Franzosa, 2019*) as well as a mouse model of inflammation (*Hughes, 2017*) with a focus on hydrogenases. Furthermore, we assessed whether hydrogenase activity enhances fitness of *Escherichia coli* in murine models of colitis. Our data suggest that utilization of molecular hydrogen contributes to the outgrowth of Enterobacteriaceae in the inflamed gut.

## Results

### Shotgun metagenomic sequencing of the murine and human gut microbiota reveals changes in bacterial hydrogen metabolism coding capacity during intestinal inflammation

To investigate $H_2$ metabolism in the inflamed gut, we reanalyzed a previously published dataset of shotgun metagenomic sequencing of cecal microbiota from a murine colitis model (*Hughes, 2017*). Cecal samples for metagenomic sequencing were collected from animals that were treated with dextran sulfate sodium (DSS) in the drinking water to induce epithelial injury and subsequent mucosal inflammation (*Chassaing, 2014*) or mock treated (*Hughes, 2017*). Comparing DSS-treated to mock-treated animals, we observed disparate abundance of various hydrogenases and their respective subunits, with enrichment of genes encoding some hydrogenases and depletion of others (*Figure 1—figure supplement 1*). This initial analysis suggested that $H_2$ metabolism might be altered during intestinal inflammation and motivated our subsequent studies.

Hydrogenase activity can be predicted based on primary sequence (*Søndergaard et al., 2016*). The HydDB database has been widely utilized to classify hydrogenases (*Dong, 2020*; *Mei, 2020*; *Panwar, 2020*; *Park et al., 2020*; *Picone et al., 2020*; *Stairs et al., 2020*; *Wong, 2020*; *Yu et al., 2020*). We confirmed the reliability of gene annotations in the HydDB database using a simulated metagenomic dataset of hydrogenase-containing and hydrogenase-free genomes (see Materials and methods for details). To evaluate the abundance of hydrogenases with different activities in the murine colitis model, reads from the metagenomic sequencing experiment were aligned to the curated HydDB hydrogenase database and segregated based on predicted activity (*Søndergaard et al., 2016*). The number of normalized reads that aligned to predicted bifurcating hydrogenases was virtually unchanged (*Figure 1A*). The relative abundance of predicted bidirectional, evolving and sensory hydrogenases decreased modestly in the DSS-treated mice; however, this difference was not statistically significant (*Figure 1A*). Notably, reads aligned to predicted uptake hydrogenases, enzymes responsible for oxidation of $H_2$ via respiration, had a significantly higher abundance in DSS-treated mice than in mock-treated mice (p<0.05) (*Figure 1A*).

We next sought to determine whether uptake hydrogenases are differentially abundant in IBD patients compared to healthy individuals. We analyzed a previously published shotgun metagenomic sequencing of stool samples from IBD patients and non-IBD controls (*Franzosa, 2019*). The stool samples used in the study by Franzosa and colleagues were collected from individuals enrolled in PRISM the Prospective Registry in IBD Study at Massachusetts General Hospital; Boston, USA (*Franzosa, 2019*) and in two studies in the Netherlands: LifeLines DEEP (*Tigchelaar et al., 2015*) and NLIBD (*Franzosa, 2019*). We aligned metagenomic reads from the aforementioned cohorts to the HydDB hydrogenase database (*Søndergaard et al., 2016*) to assess the abundance of genes encoding hydrogenases with predicted functions (*Figure 1B–C*). Mirroring our observations in the murine model of colitis, we detected a significantly higher abundance of predicted uptake hydrogenases in patients with IBD than non-IBD controls (p<0.001 for patients with Crohn's disease; p<0.01 for patients with ulcerative colitis; *Figure 1B–C*). Reads that aligned to predicted bifurcating or sensory hydrogenases were slightly more abundant in the ulcerative colitis patients (*Figure 1C*), but virtually unchanged in the Crohn's disease patients (*Figure 1B*). In contrast with our findings in the murine gut (*Figure 1A*), there was a significantly higher abundance of reads that aligned to predicted evolving hydrogenases in IBD samples (*Figure 1B–C*). These data suggest that microbial $H_2$ metabolism is altered in human IBD patients and in a mouse model of inflammation. Furthermore, the increase in relative abundance of predicted uptake hydrogenase genes during gut inflammation suggests that organisms that encode $H_2$-utilizing hydrogenases may have a fitness advantage during intestinal inflammation.

### Hydrogen utilization promotes fitness of *E. coli* in the inflamed gut

To investigate whether changes in $H_2$ metabolism contribute to the inflammation-associated expansion of Enterobacteriaceae populations, we focused on commensal *E. coli* as a representative organism. The *E. coli* K-12 genome encodes four hydrogenases: two $H_2$-oxidizing enzymes, hydrogenase-1 (Hyd-1) and hydrogenase-2 (Hyd-2), and two $H_2$-evolving enzymes (hydrogenase-3 and hydrogenase-4). Hyd-1 and Hyd-2 are encoded by the *hya* and *hyb* operons, respectively (*Figure 2A*). We therefore hypothesized that Hyd-1 and Hyd-2 might provide a fitness advantage to Enterobacteriaceae in the inflamed

**Figure 1.** Mapping of metagenomic sequencing data from murine cecal samples and human stool to bacterial hydrogenases. (**A**) Shotgun metagenomic sequencing of mock or dextran sulfate sodium (DSS)-treated mice with endogenous Enterobacteriaceae (obtained from Charles River) was previously performed to generate the analyzed dataset (ENA accession number PRJEB15095; *Hughes, 2017*). Reads were aligned to the HydDB hydrogenase database (*Søndergaard et al., 2016*) and segregated based on predicted hydrogenase activity. Each symbol corresponds to the average number of normalized reads that map to a specific sequence in the mock or DSS-treated animals (six mice per group). Averages equal to zero were assigned a value of 0.05. (**B, C**) Analysis of a previously published metagenomic sequencing dataset from stool samples collected from non-inflammatory bowel disease (IBD) controls and patients with Crohn's disease or ulcerative colitis (SRA BioProject number PRJNA400072; *Franzosa, 2019*). Reads from non-IBD controls (55 samples) and patients with Crohn's disease (87 samples) (**B**) or ulcerative colitis (76 samples) (**C**) were aligned to the HydDB hydrogenase database of predicted hydrogenase activities (*Søndergaard et al., 2016*). Each symbol corresponds to the average number of normalized reads that map to a specific hydrogenase sequence. Averages equal to zero were assigned a value of 0.0005. Medians are labeled with a red solid line, and error bars correspond to interquartile ranges. Statistical significance was determined by Bonferroni-corrected Mann–Whitney U-test (*p<0.05; **p<0.01, ***p<0.001; ns: not statistically significant). See also *Figure 1—figure supplement 1*, a *Figure 1—source data 1*, *Figure 1—source data 2*, and *Figure 1—source data 3*.

The online version of this article includes the following figure supplement(s) for figure 1:

**Source data 1.** Mapping of metagenomic sequencing of murine cecal content to hydrogenases.

**Source data 2.** Mapping of metagenomic sequencing of non-inflammatory bowel disease controls and patients with Crohn's disease to hydrogenases.

**Source data 3.** Mapping of metagenomic sequencing of non-inflammatory bowel disease controls and patients with ulcerative colitis to hydrogenases.

**Figure supplement 1.** Shotgun metagenomic sequence analysis of the cecal microbiota in the dextran sulfate sodium (DSS) colitis model.



**Figure 2.** Hydrogenases provide a competitive fitness advantage for *E. coli* during acute colitis. (**A**) Schematic representation of the hydrogenase-1 and hydrogenase-2 encoding gene loci in *E. coli* Nissle 1917 (EcN) and MP1. The DNA regions that were removed from the Δ*hya* and Δ*hyb* mutants in EcN and MP1 are indicated in black (EcN) and brown (MP1), respectively. (**B–D**) Groups of wild-type (WT) male (**C, D**) and female (**D**) C57BL/6 mice devoid of native Enterobacteriaceae were treated with 3% dextran sulfate sodium (DSS) in the drinking water. Mice were orally inoculated with a 1:1 ratio of the WT strain and an isogenic Δ*hya* Δ*hyb* mutant on day 4 of DSS treatment. (**B**) Schematic representation of the colitis model. (**C, D**) Intestinal content was collected on day 9 to determine the abundance of the EcN (**C**) or MP1 (**D**) WT strain (black bars) or isogenic Δ*hya* Δ*hyb* mutant (gray bars). The competitive index (CI) is indicated above each set of bars. Each symbol corresponds to one mouse. Bars represent geometric means ± 95% confidence intervals. Statistical significance was determined by paired Student's *t*-test of the log-transformed data (**p<0.01; ***p<0.001). See also ***Figure 2— figure supplement 1*** and 2.

The online version of this article includes the following figure supplement(s) for figure 2:

**Figure supplement 1.** In vitro analysis of *E. coli* wild-type (WT) and hydrogenase-deficient mutant strains.

**Figure supplement 2.** Assessment of bacterial fitness in the murine intestinal lumen.

gut. To determine the contribution of $H_2$ utilization, we used two commensal *E. coli* strains. Nissle 1917 (EcN) was originally isolated from a human (***Grozdanov, 2004***) and MP1 is a mouse isolate (***Lasaro, 2014***). We initially generated isogenic mutants lacking Hyd-1 and Hyd-2 activity (Δ*hya* Δ*hyb* mutant) (***Figure 2A***). As expected, inactivation of Hyd-1 and Hyd-2 activity had no discernible effect on growth of EcN under standard, aerobic laboratory conditions (***Figure 2—figure supplement 1A***). We next co-cultured the EcN wild-type and an isogenic Δ*hya* Δ*hyb* mutant under anaerobic conditions in the presence of 5% molecular hydrogen with mucin as a carbon source (***Figure 2—figure supplement***

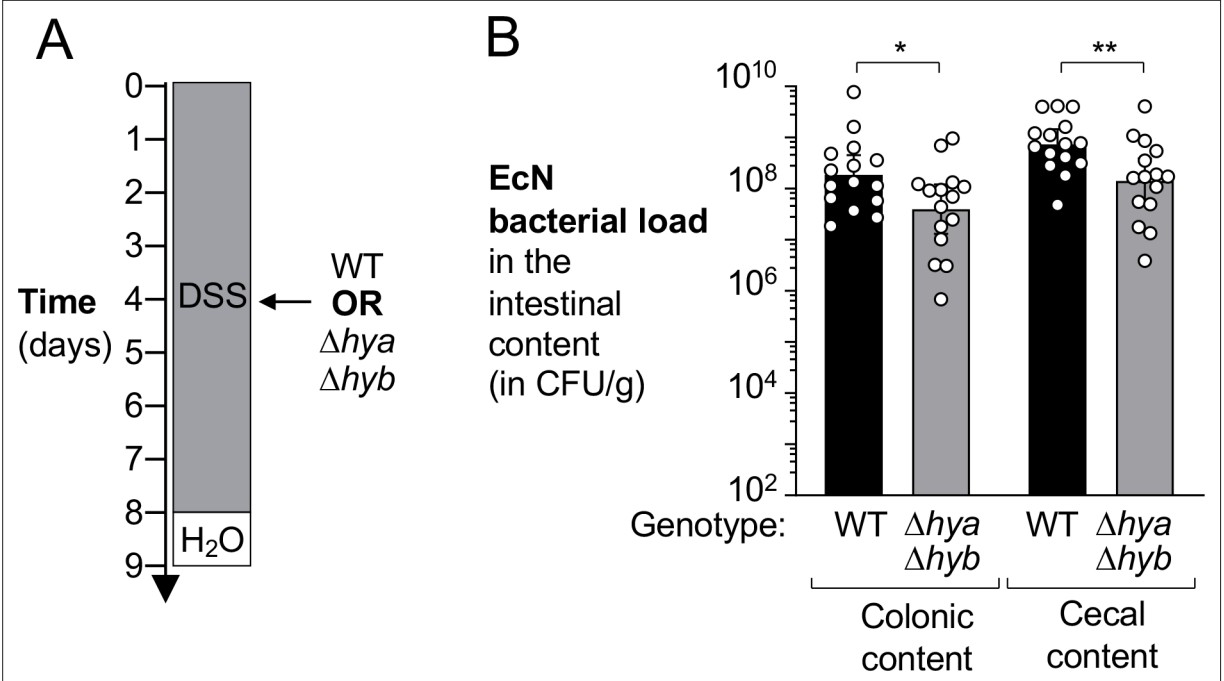

**Figure 3.** Hydrogenases enhance growth of *E. coli* in a colitis model. Wild-type (WT) female C57BL/6 mice were treated with 3% dextran sulfate sodium (DSS) in the drinking water to induce colitis. On day 4 of DSS treatment, mice were orally inoculated with either the WT *E. coli* Nissle 1917 (EcN) strain or the isogenic Δ*hya* Δ*hyb* mutant strain. (**A**) Schematic representation of the experiment. (**B**) Intestinal content was collected 9 days later to determine the abundance of *E. coli* in the colon and cecum. Each symbol corresponds to the *E. coli* bacterial abundance in one mouse. Bars represent geometric means ± 95% confidence intervals. Black bars, WT. Gray bars, Δ*hya* Δ*hyb* mutant. Statistical significance was determined by unpaired Student's *t*-test of the log-transformed data (*p<0.05; **p<0.01).

*1B*). After 18 hr, the ratio of the two strains, corrected by the corresponding ratio in the inoculum, was determined (competitive index). Consistent with previous reports (*Yamamoto and Ishimoto, 1978*), $H_2$ utilization only provided a growth advantage when an external electron acceptor, such as fumarate, was added to the media. The Hyd-2 enzyme is primarily responsible for enhancing growth under these culture conditions since a mutant lacking both Hyd-1 and Hyd-2 (Δ*hya* Δ*hyb*) was outcompeted by a strain only lacking Hyd-1 (Δ*hya*); conversely, a Δ*hyb* and a Δ*hya* Δ*hyb* mutant were equally fit in this assay (*Figure 2—figure supplement 1B*). Genetic complementation restored the phenotype of Hyd-2-deficiency (*Figure 2—figure supplement 1C*).

We next examined whether Hyd-1 and Hyd-2 contribute to *E. coli* colonization of the murine large intestine. C57BL/6 mice acquired from the Jackson Laboratory (Bar Harbor, ME) do not have any detectable endogenous Enterobacteriaceae (data not shown), which facilitates engraftment and recovery of exogenously introduced *E. coli*. To facilitate recovery of exogenously introduced *E. coli* strains, the wild-type and mutant strains were marked with low-copy number plasmids (*Wang and Kushner, 1991*; *Winter, 2013*; *Figure 2—figure supplement 2A,B*). Groups of mice were given DSS in the drinking water to induce colitis (*Figure 2B*). On day 4, coinciding with disease onset as determined by body weight loss (*Figure 2—figure supplement 2C,D*), mice were orally inoculated with an equal mixture of the respective wild-type strain and the isogenic Δ*hya* Δ*hyb* mutant strain. Five days after colonization, the abundance of each strain in the colonic and cecal content was determined and the competitive index was calculated (*Figure 2C,D*). The EcN wild-type strain outcompeted the Δ*hya* Δ*hyb* mutant in the colonic and cecal content (12-fold in the colon and 5.8-fold in the cecum; *Figure 2C*), while the MP1 wild-type strain was recovered in significantly higher numbers than the uptake hydrogenase-deficient mutant in the colonic and cecal content (25-fold and 22-fold, respectively; *Figure 2D*).

We also determined whether *hya* and *hyb* promote gut colonization in the absence of a close competitor (*Figure 3*). Mice were treated with DSS and colonized with either the EcN wild-type strain or an isogenic Δ*hya* Δ*hyb* mutant and colonization of the large intestine content assessed after 5

days (*Figure 3A*). Consistent with our observations in the competitive colonization experiments, the wild-type strain was more abundant in the colonic and cecal content than the uptake hydrogenase-deficient mutant (4.7-fold and 5.1-fold, respectively; *Figure 3B*). Collectively, these results suggest that $H_2$ utilization via Hyd-1 and Hyd-2 enhances fitness and contributes to *E. coli* colonization of the inflamed large intestine.

## Hydrogen utilization contributes to the expansion of *E. coli* during intestinal inflammation

We next wanted to assess whether $H_2$-utilizing hydrogenases contributed to inflammation-associated expansion of *E. coli* populations, where C57BL/6 mice are colonized with *E. coli* prior to induction of colitis. We chose to use the MP1 strain for this experiment as it is a mouse commensal isolate that

**Figure 4.** Hydrogenases contribute to expansion of *E. coli* in the inflamed gut. Groups of wild-type (WT) male C57BL/6 mice were orally inoculated with a 1:1 ratio of the *E. coli* MP1 WT strain and the isogenic Δ*hya* Δ*hyb* mutant. On the same day as the oral inoculation, mice received water or 3% dextran sulfate sodium (DSS) in the drinking water. (**A**) Schematic representation of experiment. (**B**) Mouse body weights. Data points represent geometric means ± standard error (mock-treated, black circles; DSS-treated, gray squares). (**C**) Hematoxylin and eosin-stained colonic and cecal sections were scored by a veterinary pathologist for submucosal edema (light gray bars), immune infiltration by polymorphonuclear cells (PMN) (dark gray bars), epithelial damage (medium gray bars), and exudate (black bars). Bars for each histopathology category correspond to the average per group. (**D, E**) The abundance of each strain was determined in the colonic (**D**) and cecal (**E**) content (WT strain, black bars; Δ*hya* Δ*hyb* mutant, gray bars). The competitive index (CI) is indicated above the sets of DSS-treated bars. Each symbol corresponds to one mouse. Dashed line corresponds to the limit of detection. Samples with values below the limit of detection were assigned a value of 10 CFU/g. Bars represent geometric means ± 95% confidence intervals. Statistical significance was determined by paired Student's *t*-test of the log-transformed data (\*\*\*p<0.001).

**Figure 5.** Hydrogenase-dependent fitness of *E. coli* correlates with intestinal inflammation. Groups of wild-type (WT) male (2–3 per group) and female (3–4 per group) C57BL/6 mice were orally inoculated with a 1:1 ratio of the *E. coli* MP1 WT strain and the isogenic Δ*hya* Δ*hyb* mutant. On the same day as the oral inoculation, mice received 3% dextran sulfate sodium in the drinking water. Groups of mice were euthanized on days 1, 3, 5, 7, and 9 after inoculation. (**A**) Mouse body weights. Data points represent geometric means ± standard error. (**B**) Colon lengths. Each symbol corresponds to data from one mouse. Bars represent geometric means ± 95% confidence intervals. (**C, D**) The competitive indices of the WT strain and the Δ*hya* Δ*hyb* mutant in the colonic (**C**) and cecal (**D**) contents were determined. Each symbol corresponds to one mouse. Bars represent geometric means ± 95% confidence intervals. Statistical significance was determined by the Kruskal–Wallis test with Dunn's *post hoc* multiple analyses test (*p<0.05; **p<0.01; ***p<0.001). See also *Figure 5—figure supplements 1–4*.

The online version of this article includes the following figure supplement(s) for figure 5:

**Figure supplement 1.** Expression of pro-inflammatory markers in time-course experiment.

**Figure supplement 2.** Development of pathological lesions during dextran sulfate sodium treatment.

**Figure supplement 3.** Hydrogenase-dependent competitive fitness of *E. coli* correlates with colon length.

**Figure supplement 4.** Hydrogenases do not provide a competitive fitness advantage to *E. coli* in healthy mice colonized for 3 days.

colonizes the mouse intestinal tract in the absence of inflammation (*Lasaro, 2014*). Groups of mice were orally inoculated with a mixture of the MP1 wild-type strain and the Δ*hya* Δ*hyb* mutant. One group received DSS treatment, while the other group was mock-treated (no inflammation) (*Figure 4A–C*). Under homeostatic conditions, the wild-type strain and the Δ*hya* Δ*hyb* mutant colonized the colon and cecum at low levels (*Figure 4D and E*). However, the wild-type strain had a marked fitness advantage over the Δ*hya* Δ*hyb* mutant in the DSS-treated mice (*Figure 4D and E*). Of note, the MP1 wild-type strain outcompeted the Δ*hya* Δ*hyb* mutant to a higher degree in this experiment than in the previous experiment in which *E. coli* strains were introduced at the onset of inflammation (*Figure 2D*).

To better understand *E. coli* $H_2$ metabolism in the context of gut inflammation, we colonized groups of DSS-treated mice with a mixture of the MP1 wild-type and the Δ*hya* Δ*hyb* mutant and acquired

samples in 2-day intervals (*Figure 5*). Disease severity and intestinal inflammation developed over time, as quantified by loss of body weight (*Figure 5A*), diminished colon length (*Figure 5B*), increased mRNA levels of pro-inflammatory cytokines (*Cxcl1* and *Tnfa*) (*Figure 5—figure supplement 1A,B*), and manifestation of pathological changes (*Figure 5—figure supplement 2A,B*). At the same time as inflammation developed, the magnitude of the phenotype conferred by $H_2$ utilization in *E. coli* increased and stayed constant at later time points (*Figure 5C–D*). Notably, colon length, a sensitive measure of colitis, was inversely correlated with the competitive fitness advantage of the wild-type strain over the Δ*hya* Δ*hyb* mutant (*Figure 5—figure supplement 3A,B*). In the absence of inflammation, the fitness of the wild-type strain and the Δ*hya* Δ*hyb* mutant were somewhat comparable at 3 days (*Figure 5—figure supplement 4*) and 9 days and after initial colonization (*Figure 4D–E*). Taken together, we conclude that Hyd-1 and Hyd-2 provide a notable fitness advantage to *E. coli* during the inflammation-associated expansion of the Enterobacteriaceae population.

**Figure 6.** Hydrogenases promote fitness of *E. coli* in piroxicam-accelerated *Il10⁻/⁻* colitis models. Groups of *Il10⁻/⁻* C57BL/6 and *Il10⁻/⁻* BALB/c mice received piroxicam-fortified diet. Mice were orally inoculated with a 1:1 ratio of the *E. coli* wild-type (WT) strain and the isogenic Δ*hya* Δ*hyb* mutant on day 2 of piroxicam treatment. Nissle 1917 (EcN) and MP1 strains of *E. coli* were used. (**A**) Schematic representations of colitis models. (**B**) Body weights. Data points represent geometric means ± standard error (EcN-colonized *Il10⁻/⁻* C57BL/6, white triangles; EcN-colonized *Il10⁻/⁻* BALB/c, black circles; MP1-colonized *Il10⁻/⁻* BALB/c, gray squares). Four mice were excluded from analysis of the MP1-colonized *Il10⁻/⁻* BALB/c mice on days 11–12 as the body weights of those mice were not available. (**C**) Abundance of the EcN WT strain and the Δ*hya* Δ*hyb* strain in the intestinal content of piroxicam-treated *Il10⁻/⁻* male C57BL/6. (**D**) Abundance of the EcN WT strain and the Δ*hya* Δ*hyb* strain in the intestinal content of piroxicam-treated *Il10⁻/⁻* male BALB/c mice. (**E**) Abundance of the MP1 WT strain and the Δ*hya* Δ*hyb* mutant in the intestinal content of piroxicam-treated *Il10⁻/⁻* male and female BALB/c. (**C–E**) CI: competitive index. Each symbol corresponds to one mouse. Dashed line corresponds to the limit of detection. Samples with values below the limit of detection were assigned a value of 10 CFU/g. Bars represent geometric means ± 95% confidence intervals. Statistical significance was determined by paired Student's *t*-test of the log-transformed data (*p<0.05; ***p<0.001).

## Hyd-1 and Hyd-2 enhance fitness of *E. coli* in the piroxicam-accelerated *Il10*[-/-] colitis model

DSS promotes mucosal inflammation by causing epithelial injury (*Cooper, 1993*). To determine whether $H_2$ utilization also provides a fitness advantage to *E. coli* in a colitis model in which inflammation was induced by a different mechanism, we used a genetic model of colitis. Conventionally raised mice deficient for the anti-inflammatory cytokine IL-10 (encoded by *Il10*) develop colitis spontaneously (*Kuhn, 1993*; *Sellon, 1998*). This process can be accelerated via oral administration of piroxicam, a nonselective nonsteroidal anti-inflammatory drug (*Berg et al., 2002*). Groups of *Il10*[-/-] mice were fed piroxicam-fortified chow instead of regular chow over the course of the experiment and were orally inoculated with an equal mixture of wild-type *E. coli* EcN or MP1 strain and the respective isogenic Δ*hya* Δ*hyb* mutants 2 days after the start of the piroxicam treatment (*Figure 6A*). We used three different experimental designs in which we varied the mouse genetic background, piroxicam dose, duration of the experiment, and *E. coli* strain used (*Figure 6A*). Mice exhibited differential susceptibility and weight loss was most prominent in the C57BL/6 background (*Figure 6B*). Importantly, the wild-type *E. coli* strains significantly outcompeted the Δ*hya* Δ*hyb* mutants, regardless of the experimental setting (*Figure 6C–E*). Taken together, $H_2$ utilization provides a fitness advantage to *E. coli* in both chemically induced and genetically induced models of murine colitis.

## Hydrogen utilization in the inflamed gut partially depends on fumarate, nitrate, and oxygen respiration

Hyd-1 and Hyd-2 are both $H_2$-utilizing, membrane-bound hydrogenases that transfer electrons from $H_2$ oxidation to the electron transport chain. In vitro, hydrogen oxidation is coupled to fumarate reduction and nitrate respiration (*Figure 2—figure supplement 1B*; *Laurinavichene and Tsygankov, 2001*; *Yamamoto and Ishimoto, 1978*). The redox potential of the nitrate/nitrite couple is more favorable than that of fumarate/succinate. As such, nitrate is the preferred electron acceptor for $H_2$ utilization in *E. coli* (*Figure 7A*).

During gut inflammation, levels of inducible nitric oxide synthase (*Nos2*) increase (*Figure 5—figure supplement 1C*; *Singer, 1996*; *Winter, 2013*) enabling nitrate respiration, which supports growth of Enterobacteriaceae family members (*Hughes, 2017*; *Winter, 2013*). In addition, *E. coli* respires oxygen using the cytochrome bd-II oxidase enzyme (AppBCX) (*Chanin, 2020*). We therefore examined whether $H_2$ utilization in the murine gut was dependent on fumarate reductase, cytochrome bd-II oxidase, or nitrate reductase activity. We assessed the fitness advantage provided by Hyd-1 and Hyd-2 in the presence (wild-type strain vs. a Δ*hya* Δ*hyb* mutant) and absence of a specific reductase (e.g., Δ*frd* mutant vs. Δ*frd* Δ*hya* Δ*hyb* mutant) in the DSS colitis model (*Figure 7B–C*). The competitive advantage conferred by Hyd-1 and Hyd-2 was significantly reduced in the absence of fumarate reductase, cytochrome bd-II oxidase, or nitrate reductase activity. Consistent with the idea that all three electron acceptors contribute to the $H_2$ utilization phenotype, inactivation of each reductase did not completely abolish the phenotype (*Figure 7B–C*).

Prior work had revealed sex-specific differences in the development of disease in the DSS colitis model (*Bábíčková et al., 2015*). When we stratified the data shown in *Figure 7B–C* according to mouse sex, we observed no striking differences in the magnitude of *E. coli* $H_2$ utilization in male and female mice (*Figure 7—figure supplement 1*).

## Hyd-1 and Hyd-2 individually contribute to fitness of *E. coli*

Hyd-1 and Hyd-2 are members of distinct [NiFe]-hydrogenase subgroups, differing in subunit composition and the range of redox potentials at which they function optimally (*Beaton et al., 2018*; *Greening, 2016*; *Lukey, 2010*; *Volbeda et al., 2013*) (reviewed in *Pinske and Sawers, 2016*). Therefore, we decided to investigate which hydrogenase mediated the fitness advantage conferred by $H_2$ utilization. In the DSS colitis model (*Figure 8A*), the EcN wild-type strain outcompeted the Δ*hya* mutant in the colonic and cecal content (3.3-fold and 2.2-fold, respectively) (*Figure 8B*). Similarly, the Δ*hyb* mutant was recovered in significantly lower numbers than the EcN wild-type strain from both colonic and cecal content (6.4-fold and 4.7-fold, respectively; *Figure 8C*). We thus conclude that both Hyd-1 and Hyd-2 provide a fitness advantage for *E. coli* during colitis.

**Figure 7.** Utilization of various electron acceptors facilitates hydrogenase-dependent competitive outgrowth of *E. coli*. (**A**) Mucin broth supplemented with or without the electron acceptors fumarate (25 mM) and nitrate (0.4 mM or 40 mM) was inoculated with an equal mixture of the indicated EcN strains. Cultures were incubated anaerobically, in the presence of 5% $H_2$, for 18 hr and the competitive index was determined. Each symbol corresponds to a biological replicate. Bars represent geometric means ± 95% confidence intervals. WT: wild-type strain; NR: *narG narZ napA* mutant. Statistical significance was determined for the indicated comparisons using ANOVA with Sidak's multiple comparisons test of the log-transformed data (**p<0.01, ***p<0.001; ns: not statistically significant). (**B, C**) Groups of WT male and female C57BL/6 mice were treated with 3% dextran sulfate sodium (DSS) in the drinking water. Mice were orally inoculated with a 1:1 ratio of the indicated *E. coli* Nissle 1917 (EcN) strains on day 4 of DSS treatment. Colonic (**B**) and cecal (**C**) content was collected after 9 days of DSS treatment to determine the competitive indices. Each symbol corresponds to one mouse. Bars represent geometric means ± 95% confidence intervals. Statistical significance was determined by the Kruskal–Wallis test with Dunn's *post hoc* multiple analyses test (*p<0.05; **p<0.01; ***p<0.001). See also *Figure 7—figure supplement 1*.

The online version of this article includes the following figure supplement(s) for figure 7:

**Figure supplement 1.** Hydrogenases confer a competitive fitness advantage to *E. coli* during colitis, independent of mouse sex.

## Discussion

Molecular hydrogen metabolism is widely utilized across microbial phyla, and hydrogenases are used by bacteria in diverse ecosystems (*Adam and Perner, 2018*; *Greening, 2016*; *Jordaan et al., 2020*; *Piché-Choquette and Constant, 2019*; *Wolf, 2016*). $H_2$ metabolism supports microbial respiration, fermentation, and carbon fixation (*Vignais and Billoud, 2007*). Recent work by Greening and

**Figure 8.** Both *hya* and *hyb* enhance fitness of *E. coli* Nissle 1917 in the inflamed gut. Groups of male (3–4 per group) and female (5 per group) wild-type (WT) C57BL/6 mice received 3% dextran sulfate sodium (DSS) in the drinking water. After 4 days of DSS treatment, the mice were orally inoculated with a 1:1 ratio of the WT *E. coli* Nissle 1917 strain and the indicated isogenic mutants. (**A**) Mouse body weights. Data points represent geometric means ± standard error. (**B**) Abundance of the WT strain (black bars) and the isogenic Hyd-1 mutant (Δ*hya* mutant; blue bars) in the intestinal content. (**C**) Abundance of the WT strain (black bars) and the isogenic Hyd-2 mutant (Δ*hyb* mutant; green bars) in the intestinal content. (**B, C**) The competitive index (CI) is indicated above each set of bars. Each symbol corresponds to one mouse. Bars represent geometric means ± 95% confidence intervals. Statistical significance was determined by paired Student's *t*-test of the log-transformed data (**p<0.01; ***p<0.001).

colleagues provides a detailed classification system of hydrogenases, enabling prediction of biological function (***Greening, 2016***; ***Søndergaard et al., 2016***). In our study, we used the HydDB classification to analyze $H_2$ metabolism in the murine and human gastrointestinal tracts, of which there is still an incomplete understanding (***Benoit et al., 2020***). Our data suggests that $H_2$ metabolism is perturbed during murine non-infectious colitis and in human IBD patients. We propose that animal models of colitis could be useful tools to probe the physiological role of $H_2$ metabolism in the gut microbiota. Our work also highlights the value of analyzing large datasets based on knowledge of enzymatic functions. The HydDB database allowed us to identify inflammation-associated changes in microbial $H_2$ metabolism in the mouse and human gut that were not obvious based on standard bioinformatic analyses of shotgun metagenomic sequencing.

$H_2$ is a key metabolite involved in cross-feeding between members of the gut microbiota (reviewed in ***Smith, 2019***). $H_2$ production can be used to dispose of electrons; $H_2$ consumption allows for the utilization of $H_2$ as a high-energy electron donor. $H_2$ metabolism supports gut colonization of methanogens, acetogens, and sulfate-reducing bacteria (***Bernalier, 1996***; ***Rey, 2013***; ***Ruaud et al., 2020***; ***Samuel, 2007***), and these microbes, in turn, prevent the accumulation of a high $H_2$ partial pressure

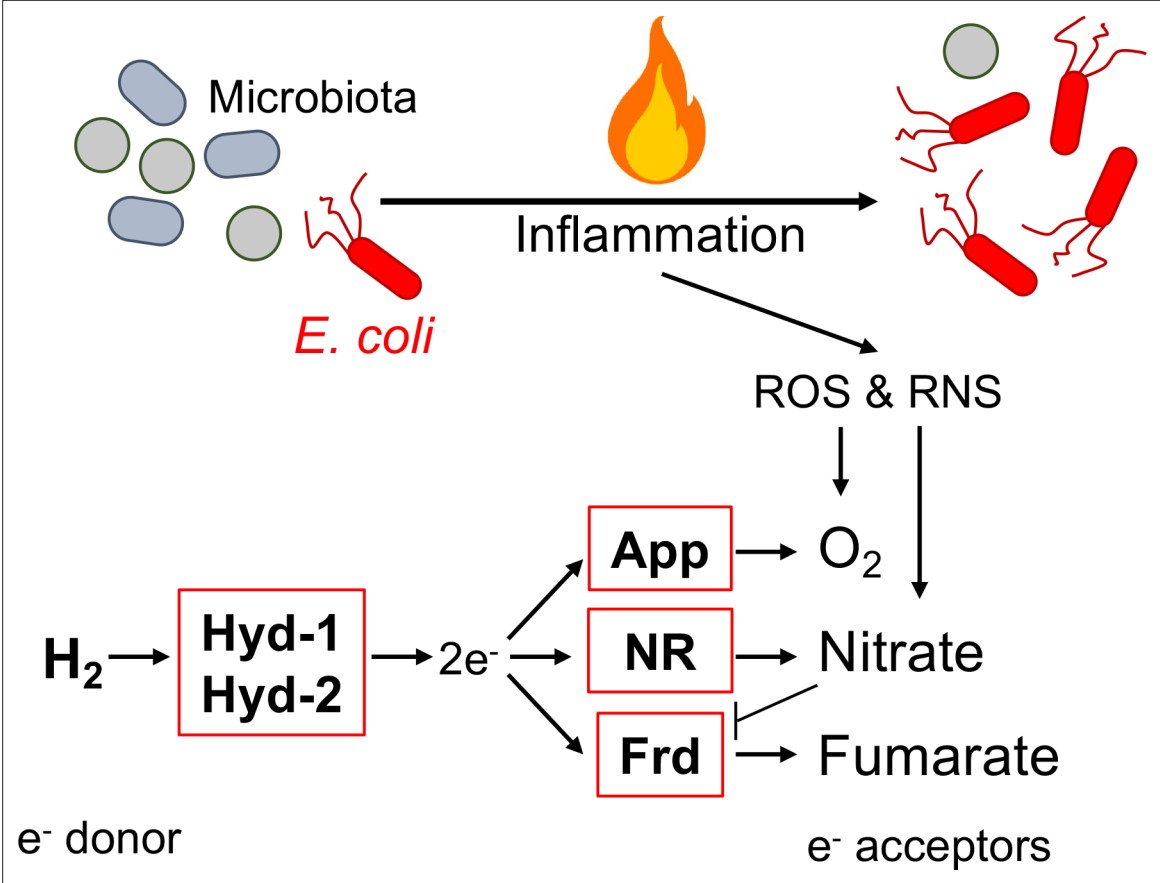

**Figure 9.** Graphical representation of findings. During inflammation, *E. coli* couples oxidation of molecular hydrogen via hydrogenase-1 (Hyd-1) and hydrogenase-2 (Hyd-2) to the reduction of oxygen, nitrate, and fumarate. Inflammatory reactive oxygen species (ROS) and reactive nitrogen species (RNS) leaking into the gut lumen allow for AppBCX (App)-mediated oxygen respiration and nitrate respiration (NR). Frd: fumarate reductase.

that can inhibit polysaccharide fermentation (*Samuel and Gordon, 2006*; *Stams, 1994*). The increased abundance of genes encoding uptake hydrogenases in microbial communities in the inflamed gut and the enhanced fitness advantage provided by $H_2$ oxidation to *E. coli* in murine models of colitis are consistent with an inflammation-disruption of $H_2$ syntrophic networks. Previous work with *Clostridium difficile* established that *C. difficile* expands due to accumulation of the metabolite succinate via loss of microbial consumption (*Ferreyra, 2014*). The lack of succinate consumption by the microbiota was shown to occur during antibiotic treatment, chemically induced motility disturbance, and impaired IL-22-mediated host glycosylation (*Ferreyra, 2014*; *Nagao-Kitamoto, 2020*). It is conceivable that changes in microbe-microbe $H_2$ exchange during inflammation might allow *E. coli* to access a $H_2$ pool.

Simple electron transport chains typically catalyze two sets of redox reactions that involve proton transfer across the membrane; in one reaction, an electron donor is oxidized, and electrons are transferred to the quinone pool; in the other reaction, electrons from the quinone pool are used to reduce a terminal electron acceptor. Here, we demonstrate that in the inflamed intestine, $H_2$ utilization via Hyd-1 and Hyd-2 requires fumarate reductase, cytochrome bd-II oxidase, or nitrate reductase activity (*Figure 9*). Nitrate is generated as a by-product of reactive nitrogen species metabolism and nitrate respiration contributes to the expansion of Enterobacteriaceae family members during flares (*Hughes, 2017*; *Winter, 2013*). Detoxification of inflammatory reactive oxygen species in the inflamed gut gives *E. coli* access to molecular oxygen and facilitates respiration via AppBCX (*Chanin, 2020*). Curiously, Hyd-1 is oxygen-tolerant (*Lukey, 2011*; *Volbeda et al., 2012*) but induced under anaerobic conditions (*Brøndsted and Atlung, 1994*; *Richard et al., 1999*). The fact that bacterial nitrate respiration and AppBCX-dependent oxygen respiration are consequences of the host's inflammatory reactive oxygen and nitrogen metabolism may in part explain why $H_2$ utilization contributes to the bloom of *E. coli* in mouse models of colitis. The concentration of free fumarate is very low in the large intestine, but

fumarate can readily be generated from other sources such as aspartate or malate (*Nguyen, 2020*; *Schubert et al., 2021*). Abolishing utilization of a single-electron acceptor did not entirely abrogate $H_2$ utilization. This outcome is consistent with a scenario in which these three electron acceptors couple to Hyd-1 or Hyd-2 in different, spatially distinct, subpopulations or at different time points as inflammation develops.

While we failed to find evidence that Hyd-1 and Hyd-2 enhance *E. coli* fitness in the absence of inflammation, it is possible that $H_2$ utilization still occurs in this setting. For example, it is conceivable that our assays were not sensitive enough to detect small fitness defects, that $H_2$ oxidation occurs in the absence of inflammation but redundant electron-donating enzymes in the electron transport chain mask the phenotype, or that *E. coli's* access to the $H_2$ pool is dependent on other microbes. Another caveat of our study is that we were not able perform functional analyses of Hyd-1 in vitro.

Hydrogenases are also widespread in enteric pathogens (*Benoit et al., 2020*) and $H_2$ metabolism contributes to gut colonization by *Helicobacter pylori* (*Olson and Maier, 2002*), *Campylobacter jejuni* (*Weerakoon, 2009*), and *Salmonella enterica* serovar Typhimurium (*S*Tm) (*Maier, 2013*). $H_2$ uptake is important for *S*Tm virulence (*Lamichhane-Khadka, 2015*; *Maier, 2004*), and it facilitates *S*Tm gut colonization (*Maier et al., 2014*; *Maier, 2013*) and fecal shedding (*Lam and Monack, 2014*). The role of hydrogenases with regards to *S*Tm systemic colonization has resulted in different outcomes, depending on the bacterial strain, route of inoculation, mouse background, and gut microbiota status (*Craig, 2013*; *Maier et al., 2014*). In our study, we observed that $H_2$ uptake contributes to gut colonization in a mouse and human commensal *E. coli* strain, and both Hyd-1 and Hyd-2 play a role in facilitating *E. coli* gut colonization during non-infectious colitis.

## Materials and methods
### Bacterial strains, plasmids, and primers
All bacterial strains and plasmids are listed in Appendix 1—key resources table. Primers are listed in Appendix 1—key resources table and *Supplementary file 1*. *E. coli* strains were routinely grown in LB broth (10 g/l tryptone, 5 g/l yeast extract, 10 g/l sodium chloride) or on LB plates (LB broth, 15 g/l agar) under aerobic conditions at 30°C or 37°C. When necessary, the antibiotics chloramphenicol (Cm), kanamycin (Kan), and carbenicillin (Carb) were added at concentrations of 15 mg/l, 100 mg/l, and 100 mg/l, respectively.

Suicide plasmids were constructed with use of a Gibson Assembly Cloning Kit (New England Biolabs, Ipswich, MA). To generate pEL1 and pEL2, regions upstream and downstream of *hyaABC* and *hybABC*, respectively, were PCR amplified from *E. coli* Nissle 1917 (EcN) with Q5 Hot Start High-Fidelity DNA Polymerase (New England Biolabs). The upstream and downstream regions of the genes of interest were inserted into SphI-digested pRDH10 by Gibson cloning. For pEL29 and pEL30, *E. coli* MP1 was used as the PCR template and the flanking regions of the genes of interest were inserted into SphI-digested pGP706. For pEL35, flanking regions of *frdABCD* were PCR amplified from EcN and inserted into SphI-digested pGP706. Prior to mutagenesis, plasmid inserts were sequenced to check for point mutations.

Suicide plasmids were propagated in DH5α $\lambda$ *pir*. S17-1 $\lambda$ *pir* was used as the donor strain to introduce suicide plasmids into EcN (pSW172) or MP1 (pSW172) strains via conjugation. Conjugation experiments were performed at 30°C to enable stable replication of the temperature-sensitive plasmid pSW172. Exconjugants in which the suicide plasmid had integrated into the chromosome were selected at 30°C with LB plates containing Cm and Carb (for the cloning with vector pRDH10) or Kan and Carb (for the cloning with vector pGP706). Mutants in which second crossover events had occurred were selected by plating on sucrose plates (5% sucrose, 15 g/l agar, 8 g/l nutrient broth base). Clean, unmarked deletions were confirmed by PCR. pSW172 was cured by growing the bacteria at 37°C. The strains EL5, EL11, EL15, EL252, EL276, EL284, EL347, EL350, and EL363 were generated using this cloning strategy.

To construct pEL32 for complementation of *hyb*, the promoter region of *hyb* and the coding sequence of *hybABC* were PCR amplified from EcN with Q5 Hot Start High-Fidelity DNA Polymerase (New England Biolabs, Ipswich, MA). The sequences of interest were inserted into EcoRI-digested pWSK129 via Gibson Assembly (New England Biolabs). pEL32 was electroporated into the appropriate EcN strain to test complementation of the *hyb* deletion.

## Mouse experiments

Specific pathogen-free (SPF) mice were used for the experiments. Male and female 6–18-week-old wild-type (WT) C57BL/6, *Il10*[-/-] C57BL/6, and *Il10*[-/-] BALB/c mice were used. Five mice at most were housed per cage. Mice were randomly assigned to groups prior to experimentation. Mice included animals obtained from the Jackson Laboratory (Bar Harbor, ME) and animals originally from the Jackson Laboratory (Bar Harbor) and bred under SPF conditions in a barrier facility at UT Southwestern. Mice were on a 12 hr light/dark cycle and had access to food and water ad libitum. All mouse experiments were reviewed and approved by the Institute of Animal Care and Use Committee at UT Southwestern.

## *E. coli* colonization experiments in the DSS-induced colitis model

Male and female WT C57BL/6 mice were used. Colitis was induced by administering a filter-sterilized solution of 3% (wt/vol) DSS (Alfa Aesar, Haverhill, MA) in water to drink. Mouse body weights and health were monitored daily. For competitive colonization experiments, mice were inoculated by oral gavage with $5 \times 10^8$ CFU of each indicated *E. coli* strain in LB broth at the indicated time points. For the single colonization experiment (*Figure 3*), mice were orally inoculated with $1 \times 10^9$ CFU of either the wild-type strain or the mutant. After 8 days of DSS treatment, DSS-supplemented water was switched to regular drinking water for 1 day. Then, mice were euthanized, and colonic and cecal contents were harvested in sterile phosphate-buffered saline (PBS; pH = 7.4) and placed on ice. To determine the abundance of the respective strains, 10-fold serial dilutions of intestinal content were plated on LB agar plates containing Kan or Carb. Wild-type and mutant strains were marked with the low-copy plasmids pWSK129 (Kan[R]) or pWSK29 (Carb[R]) to facilitate recovery from competitive colonization experiments. Colon length and colonic and cecal tissue samples were collected from the indicated experiments. Colonic and cecal tissue for quantification of mRNA was flash frozen and stored at –80°C. Colonic and cecal tissue for histopathology analysis was collected in 10% buffered formalin phosphate (Thermo Fisher) for fixation.

## *E. coli* colonization experiment in healthy mice

Male and female WT C57BL/6 mice were used. Mice were inoculated by oral gavage with $5 \times 10^8$ CFU each of the wild-type MP1 strain and the Δ*hya* Δ*hyb* mutant in LB broth. Mice were euthanized 3 days after colonization, and colonic and cecal contents were collected as described previously.

## *E. coli* colonization experiments in piroxicam-accelerated *Il10*[-/-] colitis model

Male and female 11–18-week-old *Il10*[-/-] C57BL/6 and *Il10*[-/-] BALB/c mice were used. *Il10*[-/-] C57BL/6 received piroxicam-fortified diet (100 ppm; Teklad custom research diets, Envigo, Indianapolis, IN) instead of the regular mouse chow (Teklad global 16% protein diet, irradiated, Envigo 2916) for 9 days total. *Il10*[-/-] BALB/c mice were fed piroxicam-fortified diet (50 ppm for EcN-colonized mice and 100 ppm for MP1-colonized mice) for 16 days total. The piroxicam diet was changed daily. Two days after the start of piroxicam treatment, mice were orally inoculated with $5 \times 10^8$ CFU of each indicated *E. coli* strain. At the end of the experiment, mice were euthanized and samples collected as described previously. Cages of mice in which the mice did not lose body weight or colonize with the indicated *E. coli* strains were excluded from analysis.

## Histopathology analysis

Colonic and cecal tissue was formalin-fixed (10% buffered formalin phosphate; Thermo Fisher), embedded in paraffin, and stained with hematoxylin and eosin. The samples were blinded and scored by a veterinary pathologist according to criteria described in *Winter, 2013*.

## Intestinal mRNA analysis

The relative transcription levels of *Cxcl1*, *Nos2*, and *Tnfa* genes were determined by RT-qPCR and normalized to *Gapdh* mRNA levels. RNA was extracted via the TRI reagent method (Molecular Research Center), mRNA was purified with NEBNext Poly(A) mRNA Magnetic Isolation Module (New England Biolabs), and cDNA was then synthesized with TaqMan reverse transcription reagents (Life Technologies). qPCR was performed in a QuantStudio 6 Flex Instrument (Life Technologies) with SYBR

Green (Applied Biosystems) using the primers listed in Appendix 1—key resources table. Results were analyzed using the comparative Ct method.

## Growth curves

Growth curves were performed in filter sterilized (0.22 µm) M9 minimal medium (6.8 g/l sodium phosphate dibasic anhydrous, 3 g/l potassium phosphate monobasic anhydrous, 0.5 g/l sodium chloride, 1 g/l ammonium chloride, 1 mM magnesium sulfate, 0.1 mM calcium chloride; pH = 7.0) supplemented with 20 mM glucose as the carbon source. The indicated *E. coli* strains were grown aerobically in LB broth at 37°C overnight. Then, strains were diluted in M9 medium supplemented with glucose at a final concentration of $1 \times 10^8$ CFU/ml. Cultures were incubated aerobically at 37°C with shaking at 250 rpm. The optical density at 600 nm ($OD_{600}$) was measured every 30 min. The experiment was performed in triplicate for each strain.

## Growth of *E. coli* strains in mucin broth

Porcine stomach mucin type II (100 mg) (Sigma) was sterilized by suspending mucin in 1 ml of 70% ethanol and incubating at 65°C for 2 hr. The suspension was then cooled overnight at room temperature (25°C) and the ethanol was aspirated. Mucin pellets were further dried using a vacufuge plus centrifuge (Eppendorf).

Mucin broth was generated by suspending 0.5% [w/v] dried sterile mucin in No-Carbon E medium (NCE) (3.94 g/l monopotassium phosphate, 5.9 g/l dipotassium phosphate, 4.68 g/l ammonium sodium hydrogen phosphate tetrahydrate, 2.46 g/l magnesium sulfate heptahydrate), supplemented with 1 mM magnesium sulfate. Where indicated, mucin broth was either supplemented with water, nitrate (0.4 mM or 40 mM, as indicated), or fumarate (25 mM). Overnight cultures of the indicated *E. coli* strains were used to inoculate the mucin broth at a concentration of $1 \times 10^3$ CFU/ml of each strain. Cultures were incubated anaerobically (90% $N_2$, 5% $CO_2$, 5% $H_2$; Sheldon Manufacturing) for 18 hr at 37°C in glass flasks (high surface area-to-volume ratio). Then, the abundance of the respective strains was determined by plating 10-fold serial dilutions on LB agar plates containing Kan or Carb. Wild-type and mutant strains were marked with the low-copy plasmids pWSK129 (Kan^R) or pWSK29 (Amp^R/Carb^R) to facilitate recovery from competitive growth experiments.

## Metagenomic analysis of murine samples

A published metagenomic dataset of DSS-induced murine colitis mode, available at the European Nucleotide Archive, accession number PRJEB15095 (*Hughes, 2017*), was reanalyzed to evaluate hydrogenase abundance in the cecal microbial community. Raw reads were processed using BBMap software suite (DOE Joint Genome Institute, Walnut Creek, CA) to remove adapters and low-quality reads. Reads were then decontaminated against mouse genome using Bowtie2 (*Langmead and Salzberg, 2012*). Global, untargeted mapping was performed using diamond blast (*Buchfink, 2015*) against the NCBI non-redundant database. Mapped reads were parsed, annotated, and visualized using the MEGAN5 metagenomic software suite (*Huson, 2007*; *Huson et al., 2016*).

To evaluate the abundance of different hydrogenase categories in this metagenomic dataset, diamond blast (reporting e-value cutoff: 0.001; *Buchfink, 2015*) was used to blast clean, filtered reads against the HydDB hydrogenase database (*Søndergaard et al., 2016*). Raw hits of individual samples were summarized using the FMAP_table.pl function in FMAP (*Kim, 2016*) with the -c parameter. The differential abundance of each hydrogenase category was then calculated using DESeq2 (*Love, 2014*), normalizing to the total number of reads aligned to all hydrogenases queried.

To test the accuracy of annotations in the HydDB database, we aligned simulated metagenomic datasets of hydrogenase-containing and hydrogenase-free genomes to the HydDB database and compared the relative abundance of mapped reads between datasets. Genomes of five representative members of the gut microbiome were used as a starting point. The hydrogenase-free genomes were generated by removing sequences of all known hydrogenases from the corresponding wild-type genomes. 100 bp, paired-end illumine reads were simulated using the genomes of representative members of various phyla in the gut microbiome (*Akkermansia muciniphila* ATCC BAA-835, *Bacteroides fragilis* NCTC 9343, *Faecalibacterium prausnitzii* APC918/95b, *Bacteroides thetaiotaomicron* VPI-5482, and *E. coli* Nissle 1917). The simulation was performed using ART (*Huang, 2012*) to achieve 2000-fold coverage (parameter: -f 2000) of the respective genomes. Sequences of known

hydrogenases were removed from the stated genomes and reads of the in silico knock-out genomes were simulated as stated above. The simulated reads from the wild-type and corresponding hydrogenases knock-out genomes were aligned to the HydDB database using Diamond blast with default parameters. Mapped reads were quantified using FMAP as stated above. In the simulated datasets, reads mapped to the known hydrogenases were significantly more abundant in the hydrogenase-containing dataset than in the hydrogenase-free dataset. However, 21 HydDB entries (out of 3248) had higher than twofold enrichment of reads that mapped to the hydrogenase-free dataset than the hydrogenase-containing dataset. This suggested that those 21 HydDB entries may be incorrectly annotated or may share too high of homology with non-hydrogenase encoding DNA sequences of the representative gut microbiome genomes to be reliably annotated as hydrogenases in our study. Therefore, the HydDB entries that had twofold or greater relative abundance of reads in the hydrogenase knock-out dataset than in the wild-type dataset were removed from the classification of hydrogenases to yield a curated list of mapped hydrogenases (*Figure 1—source data 1*).

### Metagenomic analysis of human samples

A published metagenomic sequencing dataset of stool samples from IBD patients and non-IBD controls (available via SRA with BioProject number PRJNA400072, *Franzosa, 2019*) was analyzed to evaluate hydrogenase abundance. Of note, of the 220 samples in this dataset, 218 samples were analyzed as the data of two samples were corrupted. Reads were trimmed and filtered against the human genome, and then aligned to the hydrogenase database (*Søndergaard et al., 2016*) using diamond blast (reporting e-value cutoff: 0.001, *Buchfink, 2015*). Raw hits of individual samples were summarized using the FMAP_table.pl function in FMAP (*Kim, 2016*) with the -c parameter. The differential abundance of each hydrogenase category was then calculated using DESeq2 (*Love, 2014*), normalizing to the total number of reads aligned to all hydrogenases queried. The HydDB entries previously excluded in the murine metagenomic analysis, based on our in silico HydDB validation, were also excluded from the human sample analysis (*Figure 1—source data 2* and *3*).

### Statistical analysis

Data were analyzed and graphs created using Microsoft Excel, PowerPoint, GraphPad Prism, and BioRender. p values <0.05 were considered statistically significant.

## Acknowledgements

We thank Dr. Mark Goulian (University of Pennsylvania) for providing *E. coli* MP1. Work in SEW's lab was funded by the NIH (AI118807, AI128151), The Welch Foundation (I-1969-20180324), the Burroughs Wellcome Fund (1017880), and a Research Scholar Grant (RSG-17-048-01-MPC) from the American Cancer Society. WZ was supported by a Research Fellows Award from the Crohn's and Colitis Foundation of America (454921). ERH and RBC were supported by a NIH T32 training grant (AI007520). CCG was supported by an NSF GRFP (1000194723). Any opinions, findings, and conclusions or recommendations expressed in this material are those of the author(s) and do not necessarily reflect the views of the funding agencies. The funders had no role in study design, data collection and interpretation, or the decision to submit the work for publication.

## Additional information

#### Competing interests

Sebastian E Winter: The corresponding author (SEW) is listed as an inventor on patent application WO2014200929A1, which describes a treatment to prevent the inflammation-associated expansion of Enterobacteriaceae.. The other authors declare that no competing interests exist.

## Funding

| Funder | Grant reference number | Author |
|---|---|---|
| National Institute of Allergy and Infectious Diseases | AI118807 | Sebastian E Winter |
| National Institute of Allergy and Infectious Diseases | AI128151 | Sebastian E Winter |
| Welch Foundation | I-1969-20180324 | Sebastian E Winter |
| Burroughs Wellcome Fund | 1017880 | Sebastian E Winter |
| American Cancer Society | RSG-17-048-01-MPC | Sebastian E Winter |
| Crohn's and Colitis Foundation | 454921 | Wenhan Zhu |
| National Institute of Allergy and Infectious Diseases | AI007520 | Elizabeth R Hughes Rachael B Chanin |
| National Science Foundation | 1000194723 | Caroline C Gillis |

The funders had no role in study design, data collection and interpretation, or the decision to submit the work for publication.

## Author contributions

Elizabeth R Hughes, Conceptualization, Investigation, writing-original-draft, Writing – review and editing; Maria G Winter, Laice Alves da Silva, Matthew K Muramatsu, Angel G Jimenez, Caroline C Gillis, Luisella Spiga, Rachael B Chanin, Renato L Santos, Investigation, Writing – review and editing; Wenhan Zhu, Conceptualization, Investigation, Writing – review and editing; Sebastian E Winter, Conceptualization, funding-acquisition, writing-original-draft, Writing – review and editing

## Author ORCIDs

Elizabeth R Hughes (iD) http://orcid.org/0000-0003-4967-8819
Sebastian E Winter (iD) http://orcid.org/0000-0003-1532-9178

## Ethics

All mouse experiments were reviewed and approved by the Institute of Animal Care and Use Committee at UT Southwestern (Protocol Numbers: 101681 and 102091). UT Southwestern uses the "Guide for the Care and Use of Laboratory Animals" when establishing animal research standards.

## Decision letter and Author response

Decision letter https://doi.org/10.7554/eLife.58609.sa1
Author response https://doi.org/10.7554/eLife.58609.sa2

# Additional files

## Supplementary files
• Supplementary file 1. Primers for mutagenesis generated for this study.
• Transparent reporting form

## Data availability

A published metagenomic dataset of DSS-induced murine colitis model (available at the European Nucleotide Archive, accession number PRJEB15095, Hughes et al., 2017) was reanalyzed to evaluate hydrogenase abundance in the cecal microbial community. A published metagenomic sequencing dataset of stool samples from IBD patients and non-IBD controls (available via SRA with BioProject number PRJNA400072, Franzosa et al., 2019) was analyzed to evaluate hydrogenase abundance. Source files have been provided for Figure1.

The following previously published datasets were used:

| Author(s) | Year | Dataset title | Dataset URL | Database and Identifier |
|---|---|---|---|---|
| Hughes ER, Winter MG, Duerkop BA, Spiga L, Furtado de Carvalho T, Zhu W, Gillis CC, Büttner L, Smoot MP, Behrendt CL, Cherry S, Santos RL, Hooper LV, Winter SE | 2016 | | https://www.ebi.ac.uk/ena/data/view/PRJEB15095 | European Nucleotide Archive, PRJEB15095 |
| Franzosa EA, Sirota-Madi A, Avila-Pacheco J, Fornelos N, Haiser HJ, Reinker S, Vatanen T, Hall AB, Mallick H, McIver LJ, Sauk JS, Wilson RG, Stevens BW, Scott JM, Pierce K, Deik AA, Bullock K, Imhann F, Porter JA, Zhernakova A, Fu J, Weersma RK, Wijmenga C, Clish CB, Vlamakis H, Huttenhower C, Xavier RJ | 2017 | | https://www.ncbi.nlm.nih.gov/bioproject/?term=PRJNA400072 | NCBI BioProject, PRJNA400072 |

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

# Appendix 1

**Appendix 1—key resources table**

| Reagent type (species) or resource | Designation | Source or reference | Identifiers | Additional information |
|---|---|---|---|---|
| Strain, strain background (*Mus musculus*) | C57BL/6 | Jackson Laboratory or bred at UT Southwestern (originally from Jackson Laboratory) | Jackson Laboratory Cat# 000664 | Wild-type |
| Genetic reagent (*M. musculus*) | *Il10⁻/⁻* C57BL/6 | Bred at UT Southwestern (originally from Jackson Laboratory) | Jackson Laboratory Cat# 002251 | B6.129P2-*Il10^{tm1Cgn}*/J |
| Genetic reagent (*M. musculus*) | *Il10⁻/⁻* BALB/c | Bred at UT Southwestern (originally from Jackson Laboratory) | Jackson Laboratory Cat# 004333 | C.129P2(B6)-*Il10^{tm1Cgn}*/J |
| Strain, strain background (*Escherichia coli*) | Nissle 1917 (EcN) | **Grozdanov, 2004** | | Wild-type strain (O6:K5:H1) |
| Strain, strain background (*E. coli*) | S17-1 $\lambda$ *pir* | **Simon et al., 1983** | | zxx::RP4 2-(Tetʳ::Mu) (Kanʳ::Tn7) $\lambda$ *pir* |
| Genetic reagent (*E. coli*) | EL5 | This study; Winter lab, UT Southwestern | | EcN Δ*hyaABC* |
| Genetic reagent (*E. coli*) | EL11 | This study; Winter lab, UT Southwestern | | EcN Δ*hybABC* |
| Genetic reagent (*E. coli*) | EL15 | This study; Winter lab, UT Southwestern | | EcN Δ*hyaABC* Δ*hybABC* |
| Genetic reagent (*E. coli*) | EL252 | This study; Winter lab, UT Southwestern | | MP1 Δ*hyaABC* |
| Genetic reagent (*E. coli*) | EL276 | This study; Winter lab, UT Southwestern | | MP1 Δ*hyaABC* Δ*hybABC* |
| Genetic reagent (*E. coli*) | EL284 | This study; Winter lab, UT Southwestern | | EcN Δ*narG* Δ*napA* Δ*narZ* Δ*hyaABC* Δ*hybABC* |
| Genetic reagent (*E. coli*) | EL347 | This study; Winter lab, UT Southwestern | | EcN Δ*frdABCD* |
| Genetic reagent (*E. coli*) | EL350 | This study; Winter lab, UT Southwestern | | EcN Δ*frdABCD* Δ*hyaABC* Δ*hybABC* |
| Genetic reagent (*E. coli*) | EL363 | This study; Winter lab, UT Southwestern | | EcN Δ*appC* Δ*hyaABC* Δ*hybABC* |
| Genetic reagent (*E. coli*) | MW139 | **Chanin, 2020** | | EcN Δ*appC* |

*Appendix 1 Continued on next page*

*Appendix 1 Continued*

| Reagent type (species) or resource | Designation | Source or reference | Identifiers | Additional information |
|---|---|---|---|---|
| Genetic reagent (*E. coli*) | SW930 | *Winter, 2013* | | EcN Δ*narG* Δ*napA* Δ*narZ* |
| Recombinant DNA reagent | pEL1 | This study; Winter lab, UT Southwestern | | Upstream and downstream regions of EcN *hyaABC* in pRDH10 |
| Recombinant DNA reagent | pEL2 | This study; Winter lab, UT Southwestern | | Upstream and downstream regions of EcN *hybABC* in pRDH10 |
| Recombinant DNA reagent | pEL29 | This study; Winter lab, UT Southwestern | | Upstream and downstream regions of MP1 *hyaABC* in pGP706 |
| Recombinant DNA reagent | pEL30 | This study; Winter lab, UT Southwestern | | Upstream and downstream regions of MP1 *hybABC* in pGP706 |
| Recombinant DNA reagent | pEL32 | This study; Winter lab, UT Southwestern | | Promoter and coding sequence of EcN *hybABC* in pWSK129 |
| Recombinant DNA reagent | pEL35 | This study; Winter lab, UT Southwestern | | Upstream and downstream regions of EcN *frdABCD* in pGP706 |
| Recombinant DNA reagent | pGP706 | *Hughes, 2017* | | *ori*(R6K) *mobRP4 sacRB* Kan^r |
| Recombinant DNA reagent | pRDH10 | *Kingsley, 1999* | | *ori*(R6K) *mobRP4 sacRB* Tet^r Cm^r |
| Recombinant DNA reagent | pSW172 | *Winter, 2013* | | *ori*(R101) *repA101ts* Carb^r |
| Recombinant DNA reagent | pSW296 | *Chanin, 2020* | | Upstream and downstream regions of EcN *appC* in pRDH10 |
| Recombinant DNA reagent | pWSK129 | *Wang and Kushner, 1991* | | *ori*(pSC101) Kan^r |
| Recombinant DNA reagent | pWSK29 | *Wang and Kushner, 1991* | | *ori*(pSC101) Carb^r |
| Sequence-based reagent | Primers used for mutagenesis | This study; Winter lab, UT Southwestern | PCR primers | Primers used for mutagenesis in this study are listed in *Supplementary file 1* |
| Sequence-based reagent | mouse *GapDH* RT-qPCR Forward Primer | *Spandidos, 2008*; *Spandidos et al., 2010*; *Wang and Seed, 2003* | Primer Bank ID 6679937a1 | AGGTCGGTGTGAACGGATTTG |
| Sequence-based reagent | Mouse *GapDH* RT-qPCR Reverse Primer | *Spandidos, 2008*; *Spandidos et al., 2010*; *Wang and Seed, 2003* | Primer Bank ID 6679937a1 | TGTAGACCATGTAGTTGAGGTCA |
| Sequence-based reagent | Mouse *Cxcl1* RT-qPCR Primer | *Godinez, 2008* | | TGCACCCAAACCGAAGTCAT |
| Sequence-based reagent | Mouse *Cxcl1* RT-qPCR Primer | *Godinez, 2008* | | TTGTCAGAAGCCAGCGTTCAC |
| Sequence-based reagent | Mouse *Nos2* RT-qPCR Primer | *Godinez, 2008* | | TTGGGTCTTGTTCACTCCACGG |

*Appendix 1 Continued on next page*

*Appendix 1 Continued*

| Reagent type (species) or resource | Designation | Source or reference | Identifiers | Additional information |
|---|---|---|---|---|
| Sequence-based reagent | Mouse *Nos2* RT-qPCR Primer | ***Godinez, 2008*** | | CCTCTTTCAGGTCACTTTGGTAGG |
| Sequence-based reagent | Mouse *Tnfa* RT-qPCR Forward Primer | ***Wilson, 2008*** | | AGCCAGGAGGGAGAACAGAAAC |
| Sequence-based reagent | mouse *Tnfa* RT-qPCR Primer | ***Wilson, 2008*** | | CCAGTGAGTGAAAGGGACAGAACC |
| Commercial assay or kit | Gibson Assembly Master Mix | New England Biolabs | Cat# E2611 | |
| Commercial assay or kit | Q5 Hot Start 2 x Master Mix | New England Biolabs | Cat# M0494 | |
| Commercial assay or kit | QIAfilter Plasmid Midi Kit | QIAGEN | Cat# 12245 | |
| Commercial assay or kit | QIAEX II Gel Extraction Kit | QIAGEN | Cat# 20021 | |
| Commercial assay or kit | TRI Reagent | Molecular Research Center | Cat# TR118 | |
| Commercial assay or kit | NEBNext Poly(A) mRNA Magnetic Isolation Module | New England Biolabs | Cat# E7490 | |
| Commercial assay or kit | TaqMan Reverse Transcription Reagents | Applied Biosystems | Cat# N8080234 | |
| Commercial assay or kit | SYBR Green qPCR Master Mix | Applied Biosystems | Cat# 4309155 | |
| Chemical compound, drug | Mucin from porcine stomach, Type II | Sigma | Cat# M2378 | Lot# SLCD8300 |
| Chemical compound, drug | Sodium nitrate | Sigma | Cat# S5506 | Lot# MKCC4317 |
| Chemical compound, drug | Sodium fumurate dibasic | Sigma | Cat# F1506 | Lot# BCCC8774 |
| Chemical compound, drug | Dextran sulfate sodium salt, MW ca 40,000 | Alfa Aesar | Cat# J63606 | Lots# T17A050, U03C023, S13C040, U01F027 |
| Chemical compound, drug | Piroxicam diet (50 ppm, 100 ppm) | Envigo | Custom diet | |
| Chemical compound, drug | LB Broth, Miller (Luria Bertani) | Becton Dickinson | Cat# 244620 | |
| Chemical compound, drug | Kanamycin sulfate | Thermo Fisher | Cat# BP906 | |
| Chemical compound, drug | Chloramphenicol | Thermo Fisher | Cat# BP904 | |

*Appendix 1 Continued on next page*

*Appendix 1 Continued*

| Reagent type (species) or resource | Designation | Source or reference | Identifiers | Additional information |
|---|---|---|---|---|
| Chemical compound, drug | Carbenicillin, disodium salt | VWR | Cat# J358 | |
| Software, algorithm | Excel | Microsoft Office | | https://www.microsoft.com/en-us/microsoft-365/excel |
| Software, algorithm | Prism v9.0 | GraphPad | | https://www.graphpad.com/scientific-software/prism/ |
| Software, algorithm | MacVector | MacVector | | https://macvector.com/ |
| Software, algorithm | QuantStudio 6 | Thermo Fisher | | https://www.thermofisher.com/us/en/home/life-science/pcr/real-time-pcr/real-time-pcr-instruments/quantstudio-systems/models/quantstudio-6-7-flex.html |
| Software, algorithm | BioRender | | | https://www.BioRender.com |
| Software, algorithm | PowerPoint | Microsoft Office | | https://www.microsoft.com/en-us/microsoft-365/powerpoint |
| Software, algorithm | BBMap software suite | Joint Genome Institute | | https://jgi.doe.gov/data-and-tools/bbtools/ |
| Software, algorithm | Bowtie 2 | *Langmead and Salzberg, 2012* | | http://bowtie-bio.sourceforge.net/bowtie2/index.shtml |
| Software, algorithm | DIAMOND | *Buchfink, 2015* | | https://github.com/bbuchfink/diamond |
| Software, algorithm | MEGAN5 | *Huson, 2007*; *Huson et al., 2016* | | https://software-ab.informatik.uni-tuebingen.de/download/megan5/welcome.html |
| Software, algorithm | FMAP | *Kim, 2016* | | https://github.com/jiwoongbio/FMAP |
| Software, algorithm | DESeq2 | *Love, 2014* | | https://bioconductor.org/packages/release/bioc/html/DESeq2.html |
| Software, algorithm | ART | *Huang, 2012* | | https://www.niehs.nih.gov/research/resources/software/biostatistics/art/index.cfm |
| Other | Anaerobic Chamber | Sheldon Manufacturing | Bactron 300 | |
| Other | European Nucleotide Archive, accession number PRJEB15095 | *Hughes, 2017* | | Metagenomic sequencing of cecal microbiota from DSS colitis mouse model |
| Other | SRA, BioProject number PRJNA400072 | *Franzosa, 2019* | | Human gut metagenome |
| Other | HydDB Hydrogenase Database | *Søndergaard et al., 2016* | | |

