## [Decision Letter]

**Acceptance summary:**

Hydrogen metabolism is important for infectious disease agents, but its role in chronic diseases like inflammatory bowel disease remain unclear. In this report, Hughes et al. analyze human microbiome data and use mouse models to demonstrate that hydrogenases are important for fitness in the inflamed gut, suggesting that hydrogen metabolism may contribute to bacterial overgrowth during colitis.

**Decision letter after peer review:**

Thank you for submitting your article "Reshaping of bacterial molecular hydrogen metabolism contributes to inflammation-associated gut microbiota dysbiosis" for consideration by *eLife*. Your article has been reviewed by 3 peer reviewers, including Peter Turnbaugh as the Reviewing Editor and Reviewer #1, and the evaluation has been overseen by Wendy Garrett as the Senior Editor.

The reviewers have discussed the reviews with one another and the Reviewing Editor has drafted this decision to help you prepare a revised submission.

Summary:

Hughes et al. investigate the role of hydrogen utilization by *E. coli* during colitis. They analyze previously published metagenomic datasets in humans and in mice to identify bacterial hydrogenases as factors that are increased during colitis. With a series of elegant in vivo experiments, by using a DSS colitis model and an IL10 colitis model, they demonstrate that hydrogenases are important for fitness in the inflamed gut for both a mouse and a human *E. coli* isolate. Overall, this is an important and well-conducted study, which nicely demonstrates the importance of hydrogen metabolism for bacterial overgrowth during colitis. The experiments are well designed and conducted, and carefully interpreted. The study fits well with the literature demonstrating a role for hydrogenases during infection with pathogens, but for the first time it shows their importance also for metabolism of non-pathogenic strains in inflammatory conditions.

Essential revisions:

1. The bioinformatics analysis is well motivated but could use a lot more work and details. The methods are unclear as to what cutoffs are used. There's no discussion of any controls that were run or prior data indicating the reliability of the hydrogenase annotations. The re-analysis of the human and mouse data did not adjust for multiple hypotheses. More importantly, gene abundance is analyzed without accounting for the increased abundance of *Enterobacteriaceae* in colitis, which could easily explain the observed gene-level enrichments. As is, I'm unconvinced that there is an association between IBD and hydrogenase abundance or the specific enrichment of "uptake hydrogenases".

2. Another major issue in my opinion is the conceptual framing around "dysbiosis", which is a problematic term due to its inconsistent use in the scientific literature but nonetheless implies some sort of overall shift in gut microbial community structure. That isn't really tested here, all of the experiments show competitive growth between artificial mutants and wild-type *E. coli*. There's also only a single experiment (Figure 4) that include healthy controls, making it impossible to determine if the expansion of *E. coli* is impacted by the loss of hya and/or hyb. Figure 4c,d shows that the double KO is still able to expand in DSS treated mice, conflicting with the hypothesis that hydrogen metabolism is required for dysbiosis. On a related note, the authors should discuss the alternative hypothesis that dysbiosis occurs prior to colitis. The assumption herein is that inflammation drives a shift in the microenvironment's redox potential which then shifts the gut microbiota. More citations are needed to explain the rationale and current evidence in support of these two alternative hypotheses in humans and mouse models. One simple experiment that could be done to address the author's dysbiosis hypothesis would be to colonize mice with a single strain at a time. Can the double KO still expand in the absence of wild-type? Does it reach a lower abundance than wild-type in mono-colonization?

3. While the genetics shows that this operon matters in vivo, there's no real data supporting whether or not hydrogenase activity is responsible, either in vitro or in vivo. Ideally additional assays and/or experiments could be added to provide support for the metabolic consequences of these deletions. At a minimum, this caveat needs to be added to the discussion and the authors should be careful not to imply that the activity matters (just that the operons do). Key controls are also missing for the bacterial genetics, including comparisons of the KO and wild-type strains during in vitro growth and complementation.

4. There is little insight into how mechanistically these hydrogenases may provide a growth advantage in the intestine, and specifically what it is about inflammation that makes Hyd-1 and Hyd-2 important? For example, how does DSS-induced weight loss change in *Enterobacteriaceae*-free (Jax B6) mice vs. those colonized with MP1 or EcN strains of *E. coli*? One might infer that EcN induces less inflammation than MP1, based upon Figure 2B vs. D, but there is no control group without *E. coli* to compare to. This leads me to the next example, which is Figure 5 piroxican Il10-/- experiments. Although the body weight of Il10-/- BALB/c mice on piroxicam is only minimally reduced (indicating less inflammation) compared to the Il10-/- C57BL/6 model, the EcN and MP1 bloom is still statistically significant. It is an oversimplification to conclude that Hyd-1 and Hyd-2 are important during "inflammation," as weight loss is the only measure of inflammation and differs between mouse strains and models. I recommend additional controls for each of these models, including DSS or piroxicam with no *E. coli* colonization, and knockout strains evaluated in the presence and absence of inflammation (i.e. no DSS to demonstrate Hyd-1 and Hyd-2 are not important under homeostatic conditions). Furthermore, there should be greater analysis into what aspect of inflammation makes Hyd-1 and Hyd-2 important for bacterial bloom – is there a direct correlation between extent of weight loss? Is there an immune cell type, cytokine signature, or histologic feature that makes Hyd-1 and Hyd-2 important? The spread of data in Figure 5 shows it would be possible to tease this apart.

5. Male and female mice were used in these experiments. DSS has a known sex difference. The authors must indicate the sex of the mice and test to see if sex could be responsible for the observed changes.

---

## [Author Response]

1. The bioinformatics analysis is well motivated but could use a lot more work and details. The methods are unclear as to what cutoffs are used. There's no discussion of any controls that were run or prior data indicating the reliability of the hydrogenase annotations. The re-analysis of the human and mouse data did not adjust for multiple hypotheses. More importantly, gene abundance is analyzed without accounting for the increased abundance of Enterobacteriaceae in colitis, which could easily explain the observed gene-level enrichments. As is, I'm unconvinced that there is an association between IBD and hydrogenase abundance or the specific enrichment of "uptake hydrogenases".

We thank the reviewers for their constructive feedback regarding the bioinformatics analysis. The reviewer raises several points, which we have addressed as follows:

We have updated the methods section to include specific cutoff values that were used.

In our analyses, the hydrogenase annotations with predicted activities were based on the HydDB database (Sondergaard et al., 2016), which has been widely utilized to classify hydrogenases (Dong et al., 2020; Mei et al., 2020; Panwar et al., 2020; Park et al., 2020; Picone et al., 2020; Stairs et al., 2020; Wong et al., 2020; Yu et al., 2020).

To assess the reliability of the hydrogenase annotations in the HydDB database, we aligned simulated metagenomic datasets of hydrogenase-containing and hydrogenase-free genomes to the HydDB database and compared the relative abundance of mapped reads between datasets. Genomes of five representative members of the gut microbiome were utilized. The hydrogenase-free genomes were generated by removing sequences of all known hydrogenases from the corresponding wild-type genomes. As predicted, in the simulated datasets, reads mapped to the known hydrogenases were significantly more abundant in the hydrogenase-containing dataset than in the hydrogenase-free dataset. However, 21 HydDB entries (out of 3,248) had more than 2-fold enrichment of reads that mapped to the hydrogenase-free dataset than in the hydrogenase-containing dataset. This data suggested that those 21 HydDB entries

may be incorrectly annotated or may share too high of homology with non-hydrogenase sequences of the representative gut microbiome genomes to be reliably annotated as hydrogenases in our study. These 21 HydDB entries were therefore removed from our analyses. We have updated the results and methods section of the revised manuscript to reflect these additional details.

Additionally, we have used Bonferroni correction to adjust for multiple hypothesis testing in our comparisons between hydrogenase activities. The new data and statistical analyses are presented in Figure 1A-C of the revised manuscript. Following exclusion of potentially incorrectly annotated hydrogenases and Bonferroni correction of the statistical analyses, the relative abundance of reads mapping to predicted uptake hydrogenases are still significantly higher in DSS-treated murine samples and human IBD patient samples than in the respective control samples. As such, these data support our interpretation that microbial H_2_ metabolism is altered during intestinal inflammation and that uptake hydrogenases may contribute to microbial fitness in the inflamed gut.

The reviewers raised a valid concern regarding our initial interpretation of the enrichment of *Enterobacteriaceae* hydrogenases in the metagenomic datasets, given the increase in *Enterobacteriaceae* known to occur during intestinal inflammation (Haberman et al., 2014; Kotlowski et al., 2007; Lupp et al., 2007). Therefore, we have chosen to remove the mapping of metagenomic sequencing reads to *Enterobacteriaceae hya* and *hyb* operons from the manuscript.

2. Another major issue in my opinion is the conceptual framing around "dysbiosis", which is a problematic term due to its inconsistent use in the scientific literature but nonetheless implies some sort of overall shift in gut microbial community structure. That isn't really tested here, all of the experiments show competitive growth between artificial mutants and wild-type *E. coli*. There's also only a single experiment (Figure 4) that include healthy controls, making it impossible to determine if the expansion of *E. coli* is impacted by the loss of hya and/or hyb. Figure 4c,d shows that the double KO is still able to expand in DSS treated mice, conflicting with the hypothesis that hydrogen metabolism is required for dysbiosis. On a related note, the authors should discuss the alternative hypothesis that dysbiosis occurs prior to colitis. The assumption herein is that inflammation drives a shift in the microenvironment's redox potential which then shifts the gut microbiota. More citations are needed to explain the rationale and current evidence in support of these two alternative hypotheses in humans and mouse models. One simple experiment that could be done to address the author's dysbiosis hypothesis would be to colonize mice with a single strain at a time. Can the double KO still expand in the absence of wild-type? Does it reach a lower abundance than wild-type in mono-colonization?

The reviewers raised important points regarding the use of the term “dysbiosis”. Generally, it describes changes in the composition and/or function and/or location of microbial communities during disease. In our study, we only carefully investigated *E. coli* metabolism in the context of inflammation, as the reviewer pointed out. We have therefore rephrased the title, parts of the abstract, and parts of the introduction, making a careful distinction between disease-associated microbiota changes and expansion of *Enterobacteriaceae* family members such as *E. coli* during inflammatory flares. The new title reads “Reshaping of bacterial molecular hydrogen metabolism contributes to the outgrowth of commensal *E. coli* during gut inflammation”.

As the reviewers have mentioned, it is unclear whether microbiota changes occur as a consequence of colitis or vice versa. Current evidence in the literature supports both hypotheses, which are of course not mutually exclusive (Chanin et al., 2020; David et al., 2014; Garrett et al., 2007; Moayyedi et al., 2015; Seregin et al., 2017; Winter et al., 2013; Zhu et al., 2018). As recommended by the reviewers, we have updated the introduction to include additional citations and a statement of hypotheses regarding the order of occurrence of “dysbiosis” and colitis. Of note, these various hypotheses highlight the need for additional studies of the mechanisms responsible for microbiota changes during inflammation and its impact on host physiology.

The reviewers make a valid point that additional competitive growth experiments in healthy animals would provide strong support for the role of inflammation in hydrogen-dependent outgrowth of *Enterobacteriaceae*. To address this concern, we performed three experiments.

1. We assessed the fitness of the wild-type strain compared to the hydrogenase-deficient strain in a time-course experiment (Figure 5). The fitness advantage conferred by hydrogen utilization developed with the onset of inflammation (Figure 5) and correlated with colon length, a sensitive marker of inflammation (Figure 5—figure supplement 3).

2. Experimentally introduced *E. coli* colonizes healthy animals inconsistently*,* making such studies challenging since, at later time points, few animals are colonized by *E. coli*. We therefore colonized healthy animals with our wild-type and hydrogenase-deficient strains for a short period of time (3 days) and recovered both strains in most animals . In the absence of inflammation, we observed similar levels of colonization by the wild-type and hydrogenase-deficient strain in these animals. This new data is included in Figure 5—figure supplement 4 of the revised manuscript.

3. We colonized mice with a single strain at a time, as suggested by the reviewers, to determine whether the fitness advantage conferred by hydrogen utilization contributes to colonization of the murine gut. We observed that the hydrogenase-deficient strain colonized DSS-treated mice to a lower level than the wild-type strain, providing further support for the idea that hydrogen utilization contributes to the outgrowth by *E. coli* during colitis. This new data is in Figure 3 of the revised manuscript.

3. While the genetics shows that this operon matters in vivo, there's no real data supporting whether or not hydrogenase activity is responsible, either in vitro or in vivo. Ideally additional assays and/or experiments could be added to provide support for the metabolic consequences of these deletions. At a minimum, this caveat needs to be added to the discussion and the authors should be careful not to imply that the activity matters (just that the operons do). Key controls are also missing for the bacterial genetics, including comparisons of the KO and wild-type strains during in vitro growth and complementation.

We have analyzed growth of our *E. coli* strains under defined laboratory conditions, as suggested. Growth of the wild-type strain and hydrogenase-deficient strains under atmospheric air conditions is virtually identical (Figure 2—figure supplement 1A), suggesting that the lack of Hyd-1/2 hydrogenase activity does not impede growth non-specifically. Consistent with previous findings (Laurinavichene and Tsygankov, 2001; Yamamoto and Ishimoto, 1978), we noted that hydrogen utilization enhanced growth under anaerobic conditions in the presence of fumarate and nitrate as the exogenous electron acceptors (Figure 2—figure supplement 1B). The phenotype in vitro is solely due to Hyd-2 activity (Figure 2—figure supplement 1B). Genetic complementation of the Hyd-2-deficient strain (native promoter and coding sequences on a low-copy number plasmid) rescues the fitness phenotype (Figure 2—figure supplement 1C). Unfortunately, we were unable to establish a phenotype in vitro for Hyd-1, presumably due to inadequate modeling of the mouse intestinal tract in vitro (Figure 2—figure supplement 1B). We have therefore edited the text to discuss the caveat that our study does not analyze the activity of Hyd-1 in vitro.

4. There is little insight into how mechanistically these hydrogenases may provide a growth advantage in the intestine, and specifically what it is about inflammation that makes Hyd-1 and Hyd-2 important? For example, how does DSS-induced weight loss change in Enterobacteriaceae-free (Jax B6) mice vs. those colonized with MP1 or EcN strains of *E. coli*? One might infer that EcN induces less inflammation than MP1, based upon Figure 2B vs. D, but there is no control group without *E. coli* to compare to. This leads me to the next example, which is Figure 5 piroxican Il10-/- experiments. Although the body weight of Il10-/- BALB/c mice on piroxicam is only minimally reduced (indicating less inflammation) compared to the Il10-/- C57BL/6 model, the EcN and MP1 bloom is still statistically significant. It is an oversimplification to conclude that Hyd-1 and Hyd-2 are important during "inflammation," as weight loss is the only measure of inflammation and differs between mouse strains and models. I recommend additional controls for each of these models, including DSS or piroxicam with no *E. coli* colonization, and knockout strains evaluated in the presence and absence of inflammation (i.e. no DSS to demonstrate Hyd-1 and Hyd-2 are not important under homeostatic conditions). Furthermore, there should be greater analysis into what aspect of inflammation makes Hyd-1 and Hyd-2 important for bacterial bloom – is there a direct correlation between extent of weight loss? Is there an immune cell type, cytokine signature, or histologic feature that makes Hyd-1 and Hyd-2 important? The spread of data in Figure 5 shows it would be possible to tease this apart.

As suggested, we analyzed what specifically about inflammation facilitates *E. coli* outgrowth via Hyd-1 and Hyd-2. Hyd-1 and Hyd-2 oxidize H_2_, thus providing electrons that can be used in *E. coli*’s electron transport chain. For example, hydrogen oxidation can be coupled to fumarate or nitrate respiration in vitro, with nitrate being the preferred electron acceptor due to its more favorable redox potential (Figure 7A). During intestinal inflammation, electron donors such as nitrate and oxygen become available. Nitrate is generated through the decay of reactive nitrogen species. Oxygen tension in the gut lumen is increased as a result of diffusion of oxygen from the vasculature into the lumen (Cevallos et al., 2019; Chanin et al., 2020; Hughes et al., 2017; Winter et al., 2013). Therefore, we tested whether nitrate reductases or a cytochrome-bd II oxidase facilitate the fitness advantage provided by Hyd-1 and Hyd-2 (Figure 7B, C). The competitive advantage conferred by Hyd-1 and Hyd-2 was significantly reduced in the absence of fumarate reductase, cytochrome bd-II oxidase, or nitrate reductase activity. Consistent with the idea that all three electron acceptors contribute to the hydrogen utilization phenotype, inactivation of each reductase did not completely abolish the phenotype. We therefore conclude that the change in availability of electron acceptors during intestinal inflammation may be a driver for hydrogenase-dependent outgrowth of *E. coli*. Of note, this new data does not exclude the possibility that disruptions in microbe-microbe H_2_ exchange between other commensal microbes may also contribute to the utility of Hyd-1 and Hyd-2 during colitis. This new data is presented in Figure 7A-C and we have updated the results and discussion.

The reviewers raised the point that our study focused primarily on body weights as a metric for intestinal inflammation. We performed a time-course experiment to address the concerns raised in major points #2 and #4, regarding our interpretation that hydrogenases provided a fitness advantage in the context of inflammation. In addition to assessing changes in body weight in this experiment, we also measured colon lengths, assessed mRNA levels of markers of inflammation in colonic and cecal tissue, and assessed inflammation in a semi-quantitative manner by histology (Figure 5—figure supplements 1 and 2). Histopathology analysis was also performed on another key experiment in the original manuscript during which competitive indices were compared between mock- and DSS- treated animals. The new data is shown in Figures 4C and Figure 5—figure supplement 1-2.

The reviewers asked whether there are differences in disease outcome between *Enterobacteriaceae*-free animals and those colonized with various *E. coli* strains. The literature suggests *Enterobacteriaceae* worsen disease outcome in animal models of IBD (Chassaing et al., 2014; Ellermann et al., 2019; Garrett et al., 2010; Zhu et al., 2018), but further mechanistic studies are needed. However, our current study does not attempt to answer such questions, and we have clarified the introduction to avoid any confusion.

5. Male and female mice were used in these experiments. DSS has a known sex difference. The authors must indicate the sex of the mice and test to see if sex could be responsible for the observed changes.

We agree with the reviewers that sex should be considered in this study. We have updated the manuscript figure legends to include what sexes were used. We repeated various experiments from the original manuscript with more female and/or male mice. We did not observe a contribution of mouse sex to the competitive advantage of the wild-type strain compared to the mutant strain lacking *hya* and *hyb*, which stratifies the data from Figure 7B-C in the revised manuscript (WT vs *hya hyb*) by mouse sex. No statistically significant differences in the magnitude of the phenotype were noted. We have included this data in Figure 7—figure supplement 1 in the revised manuscript.